# STAR: Spatio-Temporal Attention-guided Recurrence Prediction for Colorectal Cancer Liver Metastasis

## Abstract

Postoperative recurrence remains a primary obstacle to long-term survival for patients with colorectal cancer liver metastasis (CRLM).However,a major limitation of current prognostic methods is their static nature, which fails to account for the dynamic, time-evolving risk of recurrence driven by complex postoperative processes like liver regeneration and microenvironmental shifts. To overcome this challenge, we therefore developed STAR, a novel deep learning framework designed for dynamic prediction of postoperative recurrence and survival. Specifically, the model's primary innovation is its ability to forecast long-term, year-by-year prognosis by analyzing the temporal evolution of postoperative CT scans in conjunction with key clinical data. Additionally, the framework can simultaneously generate personalized, annual recurrence risk heatmaps. These heatmaps offer an intuitive, visual guide to the probable location and timing of recurrence, thereby providing clinicians with interpretable, patient-specific insights to tailor dynamic surveillance strategies. When evaluated on the MSKCC CRLM dataset, our model demonstrated outstanding predictive performance, achieving 90% accuracy in assessing survival status for each of the 12 postoperative years (Temporal Adjacency Accuracy, TAA) with a Mean Absolute Error (MAE) of 0.7500. This study consequently establishes a new paradigm for the postoperative management of CRLM, shifting the focus from static, single-point assessment to continuous, dynamic risk monitoring. Ultimately, by providing a tool for more precise and personalized follow-up, our framework helps advance the clinical goal from simple "spatial resection" to a more comprehensive "spatiotemporal cure."

## 1 Introduction

Colorectal cancer (CRC) ranks as the third most prevalent malignancy and the second principal cause of cancer-related death worldwide (Ferlay et al., 2015; Bray et al., 2018). A significant number of these patients, approximately 50%, will develop colorectal liver metastasis (CRLM), a condition where the cancer spreads to the liver. For these individuals, hepatic resection, often paired with neoadjuvant chemotherapy, represents the only potentially curative treatment option for achieving long-term survival (Zarour et al., 2017; Filip et al., 2020). Despite this treatment, the prognosis remains grim; 60–70% of patients suffer from postoperative recurrence, culminating in a 5-year overall survival rate below 30% (Filip et al., 2020; Rada et al., 2021). This critical clinical challenge is rooted in the high heterogeneity of CRLM tumors and the complex (Rada et al., 2021; Du et al., 2020), dynamic biological processes occurring after surgery, such as chemotherapy-induced steatosis and postoperative liver regeneration (Rada et al., 2021; Li et al., 2024). Consequently, conventional prognostic tools like the Fong Clinical Risk Score, which provide a single, static risk assessment, are inadequately equipped to capture the nonlinear temporal evolution of postoperative recurrence risk (Lee et al., 2023; Kim et al., 2019).

Traditional prognostic methods face significant hurdles in providing accurate, long-term predictions for CRLM patients. These models primarily depend on clinicopathological features and static imaging assessments, assuming a linear risk progression that fails to capture the complex, time-dependent interactions between postoperative liver regeneration, chemotherapy effects, and tumor biology. This leaves clinicians with an incomplete and static view of a patient's evolving risk profile

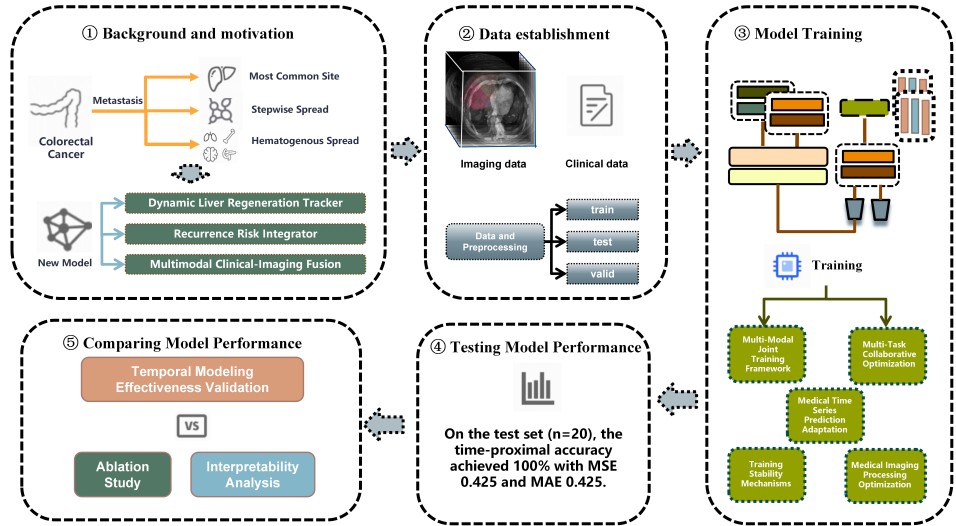

Figure 1: In this experiment, the motivation is determined through research, the dataset is constructed, the model is trained, and finally the superiority of the method is verified through evaluation and experiment.

(Rada et al., 2021). In contrast, computer-aided methods using deep learning offer powerful advantages by analyzing complex medical data to uncover deeper insights. For instance, radiomics can quantify tumor heterogeneity from CT images, but it often misses subtle, time-evolving changes in the surrounding liver tissue that are crucial for predicting recurrence (Li et al., 2024). While models incorporating clinical markers like CEA can enhance diagnostic specificity, they lack the spatial resolution to identify precisely where in the liver recurrence is most likely to emerge (Yu et al., 2024). Critically, most current advanced models are limited by their reliance on a single, preoperative snapshot, thereby overlooking the decisive influence of the postoperative period, where liver and microenvironment remodeling profoundly shape the patient's risk trajectory (Rada et al., 2021).

Our research aims to fundamentally shift the paradigm of postoperative CRLM management from a single, static prediction to dynamic, personalized monitoring by developing computer-aided automated methods. The central clinical objective is to equip clinicians with a tool that generates personalized, interannual heatmaps visualizing the evolving risk of recurrence across different regions of the liver. Such a dynamic, spatiotemporal decision-support tool can enable the formulation of tailored follow-up surveillance strategies, facilitate the earlier detection of recurrent disease, and ultimately improve patient outcomes. To achieve this, we propose STAR, an innovative deep learning architecture based on Clinical-Imaging Joint Modeling. This framework explicitly models the fourth dimension—time—by simulating and analyzing year-by-year changes in postoperative CT scans. This allows it to capture the dynamic interplay between liver regeneration and microenvironment remodeling. Clinically, this model integrates longitudinal imaging surveillance with patient data to construct a comprehensive, multimodal risk profile. By generating intuitive, yearly risk heatmaps, our work establishes an interpretable spatiotemporal decision paradigm for CRLM and supports more personalized interventions, aiming to enhance the postoperative management of patients with colorectal liver metastases (CRLM) and transform postoperative care.

## 2 RELATED WORK

In the field of colorectal cancer liver metastasis (CRLM) prognosis research, traditional prognostic models have long been constrained by their inherent limitations. These models predominantly rely on clinicopathological characteristics (e.g., CEA levels, number of liver metastases) and static imaging assessments but fail to incorporate dynamic modeling capabilities. As highlighted by (Brudvik et al., 2017) in their systematic review, conventional models based on clinicopathological factors struggle to achieve precise predictions due to their neglect of the dynamic evolution of tumor bi-

ology and nonlinear associations between genetic mutations and prognosis. Furthermore, widely adopted Cox proportional hazards models (Andersen & Gill, 1982), grounded in linear risk assumptions, cannot characterize the temporal interactions between postoperative liver regeneration and neoadjuvant chemotherapy (Rizopoulos, 2012). These shortcomings have motivated researchers to explore novel methodologies for improving prognostic accuracy.

Recent advances in integrating CT/MRI radiomics with deep learning techniques have provided new avenues for quantifying tumor heterogeneity. For instance, (Tan et al., 2025) developed a multimodal deep learning framework that enables tumor heterogeneity quantification without requiring manual segmentation. (Zhou et al., 2024) extended the application of 3D convolutional networks by proposing an integrated framework based on multiscale feature extraction. By leveraging radiomic data from CT/MRI to construct prognostic models and employing adaptive receptive field adjustment strategies to enhance spatial resolution, their approach achieved improved accuracy in assessing tumor heterogeneity and microenvironmental changes (Zhou et al., 2024). However, most existing methods focus on preoperative static imaging features and inadequately model the spatiotemporal correlations of postoperative liver regeneration and dynamic microenvironment remodeling.

Subsequently, spatiotemporal dynamic modeling has gained attention in medical imaging, yet its applicability to CRLM remains challenging. For instance, (Guo et al., 2024) proposed a trimodal temporal fusion framework validated in surgical scene graph generation tasks, and (Bastiancich et al., 2024) analyzed tumor microenvironment complexity through multiparametric imaging. More recently, the STG framework (Spatiotemporal Graph Neural Network with Fusion and Spatiotemporal Decoupling Learning) also attempted CRLM prognosis prediction, but its graph construction and decoupling design yielded limited accuracy and robustness on clinical datasets. However, these approaches, not optimized for CRLM-specific biological processes, remain difficult to directly apply to prognosis prediction. To address these limitations, we designed a lightweight 4D attention mechanism capable of precisely modeling the interannual evolution of postoperative processes such as liver regeneration and steatosis, which both reduces parameters and enhances dynamic perception of CRLM-specific biology.

In multimodal data fusion, dual challenges of spatiotemporal scale mismatch and redundant interference persist. (Litjens et al., 2017) identified that temporal-spatial disparities between clinical indicators (e.g., CEA) and imaging features (e.g., sinusoidal dilatation) may induce modality mismatch risks. For example, (Tixier et al., 2011) demonstrated that irinotecan therapy increases the coefficient of variation in CT texture features, exacerbating cross-modal feature alignment bias. (Parmar et al., 2014) compared early and late fusion strategies, concluding that neither can resolve the regulatory mechanisms of microenvironmental dynamics on imaging phenotypes. Consequently, developing causality-driven fusion models is critical. Our study proposes a Transformer-based cross-modal alignment strategy that jointly optimizes modality alignment loss and disentanglement loss functions, suppressing redundant feature interference and establishing chemotherapy response-related dynamic biomarkers. This approach provides a novel theoretical framework for multimodal dynamic modeling. These advancements not only offer new insights and methodologies for postoperative CRLM prognosis prediction but also lay a robust foundation for future research directions, driving further progress in the field.

## 3 METHODS

The proposed CRLM prognostic prediction model addresses critical clinical challenges through three integrated modules: 1) dynamic postoperative evolution tracking, 2) multimodal clinical-imaging fusion, and 3) personalized prognostic prediction (Figure 2).

### 3.1 DYNAMIC LIVER REGENERATION TRACKER

Accurate prognosis for colorectal liver metastases (CRLM) requires capturing the liver's dynamic biological evolution after surgery. Static preoperative models cannot account for critical temporal processes like liver regeneration and chemotherapy-induced steatosis, which directly impact recurrence risk. Our Dynamic Liver Regeneration Tracker addresses this by continuously modeling yearly liver changes for up to 12 years post-operation.

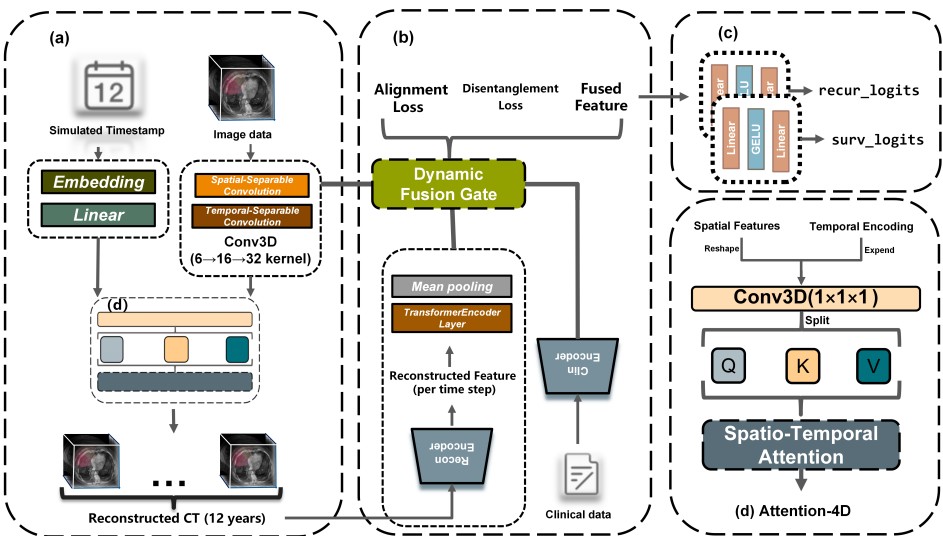

Figure 2: CRLM prognostic prediction framework based on multimodal spatiotemporal joint modeling.

The system embeds a "virtual timestamp" into each annual 3D CT scan, creating a longitudinal 4D dataset. Using efficient spatiotemporal separable convolutions, it decouples spatial analysis (tumor morphology, vascular structures) from temporal dynamics (regeneration rates, steatosis progression). This approach reduces computational complexity by 41% versus standard 3D methods while enabling long-term monitoring.

In the first stage, discrete timesteps $t \in \{1, \ldots, 12\}$ (corresponding to postoperative years) are embedded into continuous vectors $E_t$ through an embedding layer. These vectors are then broadcast and spatially aligned with spatial feature maps to form spatiotemporally fused inputs $X_t = [X_{ct}; E_t]$. In the second stage, spatiotemporal separable convolutions are employed to decouple spatial and temporal features. Specifically, a $3 \times 3 \times 3$ convolutional kernel captures local spatial features such as tumor morphology and vascular topology, while a $3 \times 1 \times 1$ convolutional kernel models microenvironmental evolution between adjacent timesteps. This design reduces parameter count by 41% compared to traditional 3D convolutions while retaining temporal modeling capability. In the third stage, 4D attention weights are computed. Spatiotemporal features and temporal encodings are concatenated along the channel dimension to generate query ($Q$), key ($K$), and value ($V$) vectors. The attention scores are then calculated as follows:

$$\text{Attention}(Q, K, V) = \text{softmax}\left(\frac{QK^T}{\sqrt{d_k}}\right) V \tag{1}$$

where $d_k$ represents the dimension of the key vectors. Based on the computed attention weights, features from each timestep are adaptively aggregated through weighted summation. Subsequently, transposed convolution is applied to restore spatial resolution, generating reconstructed feature maps that encode dynamic postoperative evolution. Finally, the aggregated features across all timesteps undergo deconvolution-based spatial upsampling, yielding high-resolution reconstructed images that reflect spatiotemporally integrated prognostic patterns.

The core of the tracker is a 4D attention mechanism that adaptively focuses on the most prognostically significant changes over time. By calculating attention weights, it learns to prioritize subtle but critical indicators of recurrence, whether they are related to insufficient regeneration, changes at the surgical margin, or evolving steatosis. Finally, the tracker synthesizes this information to generate a series of reconstructed maps, that forms a robust basis for accurate, time-aware risk prediction.

## 3.2 RECURRENCE RISK INTEGRATOR

Accurate prognosis for colorectal liver metastases requires integrating heterogeneous data streams, particularly high-resolution imaging and clinical biomarkers like serum CEA levels. Our Recur-

---

**Algorithm 1** Multimodal Clinical–Imaging Fusion

---

**Input**: Raw imaging features $\mathbf{h}_{\text{img}}$, clinical data $\mathbf{x}_{\text{clin}}$, reconstructed images $\mathbf{X}_{\text{recon}}$
**Output**: Fused features $\mathbf{h}_{\text{fused}}$, alignment loss $\mathcal{L}_{\text{align}}$, disentanglement loss $\mathcal{L}_{\text{dis}}$

1: **Encode Clinical Data:**
2: $\mathbf{h}_{\text{clin}} \leftarrow \text{ClinEncoder}(\mathbf{x}_{\text{clin}})$
3: **Extract Reconstructed Features:**
4: **for** $t = 1$ **to** $T$ **do**
5: $\quad \mathbf{h}_{\text{recon}}^{(t)} \leftarrow \text{ReconEncoder}(\mathbf{X}_{\text{recon}}^{(t)})$
6: **end for**
7: $\mathbf{h}_{\text{recon}} \leftarrow \text{Transformer}(\{\mathbf{h}_{\text{recon}}^{(t)}\}_{t=1}^{T})$
8: **Compute Alignment Loss:**
9: $\mathcal{L}_{\text{align}} \leftarrow -\dfrac{\text{mean}\big(\cos(\mathbf{h}_{\text{img}},\mathbf{h}_{\text{clin}})\big)}{\text{mean}\big(\cos(\mathbf{h}_{\text{img}},\mathbf{h}_{\text{recon}})\big)}$
10: **Compute Disentanglement Loss:**
11: $\mathcal{L}_{\text{dis}} \leftarrow \| \text{Cov}(\mathbf{h}_{\text{img}}) - \text{Cov}(\mathbf{h}_{\text{clin}})\|_F / d^2$
12: **Dynamic Fusion:**
13: $\mathbf{h}_{\text{fused}} \leftarrow \sigma(\mathbf{gate}_0)\,\mathbf{h}_{\text{img}} + \sigma(\mathbf{gate}_1)\,\mathbf{h}_{\text{recon}} + \mathbf{h}_{\text{clin}}$

---

rence Risk Integrator (Figure 2b) combines these modalities to enhance prediction of postoperative recurrence. This module transforms raw clinical inputs into noise-filtered prognostic representations $C_{\text{enc}}$ that preserve critical biomarkers while eliminating irrelevant variations. Simultaneously, time-aware CT reconstructions from the Dynamic Liver Regeneration Tracker are processed to extract global temporal features $F_{\text{time}}$, which merge with spatiotemporal features $F_{\text{spatio-temp}}$ capturing the liver's evolving pathophysiology.

To harmonize these distinct data sources, we implement a Transformer-based architecture with dual-path optimization. This approach enforces clinical-imaging consistency through specialized alignment mechanisms while preserving modality-specific information via covariance minimization. These complementary strategies ensure radiological findings corroborate blood-based biomarkers like CEA trends while retaining unique prognostic signals from each data source—whether regeneration patterns on imaging or biochemical markers in clinical records.

The integration culminates in a learnable gating mechanism that dynamically weights contributions from spatiotemporal imaging features ($F_{\text{spatio-temp}}$), temporal reconstructions ($F_{\text{time}}$), and clinical inputs ($C_{\text{enc}}$). This adaptive weighting autonomously prioritizes the most predictive elements—such as surgical margin changes on CT versus rising CEA levels—optimizing the fused representation for individualized risk assessment. Implementation details are formalized in Algorithm 1 and visualized in Figure 3.

### 3.3 Personalized Prognosis Projector

The fused spatiotemporal features drive our Personalized Prognosis Projector (Figure 2c) to generate individualized, time-sensitive risk assessments. Rather than providing a static risk score, this module projects detailed year-by-year forecasts of both cancer recurrence and overall survival probabilities throughout the critical 12-year postoperative window.

A multi-task architecture enables comprehensive prognostic modeling through parallel processing streams. The recurrence classification head employs a fully connected network to predict tumor recurrence probability for each postoperative year, while simultaneously, the survival regression head estimates corresponding annual survival probabilities. This dual-branch design allows the model to identify shared prognostic patterns influencing both outcomes while capturing task-specific clinical nuances.

By simultaneously training both prediction tasks, the projector delivers a unified yet nuanced prognostic profile that directly informs clinical decision-making. The resulting year-specific risk trajectories enable clinicians to personalize follow-up schedules and surveillance strategies according to each patient's evolving postoperative risk landscape.

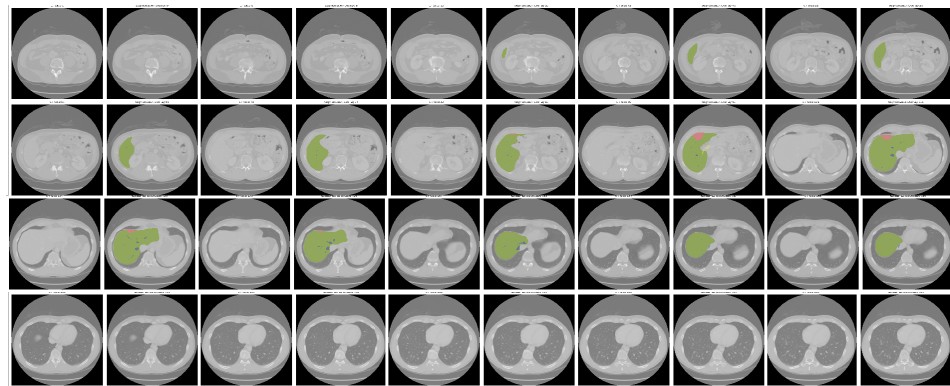

Figure 3: The dataset used in the experiment.

## 4 EXPERIMENTS

In this section, we validate the proposed multimodal spatiotemporal joint modeling framework via experiments: first, by comparing performance with a 4D spatiotemporal attention baseline; then via ablation studies to verify contributions of each module.

**Datasets.** This study is based on a publicly available dataset from the Memorial Sloan Kettering Cancer Center (MSKCC), comprising multimodal data from 197 pathologically confirmed colorectal cancer liver metastasis (CRLM) patientsSimpson et al. (2024) . Preoperative portal venous phase contrast-enhanced CT images for all patients were stored in DICOM format, covering the entire liver's 3D anatomical structure (resolution: $512\times512\times N$). Segmentation labels (liver, future liver remnant, hepatic veins, portal veins, and tumor regions) were annotated by three imaging experts. Clinical data included age, sex, primary tumor TNM staging, distribution of liver metastases, serum CEA levels, and treatment regimens. The dataset was divided into training (157), validation (20), and test sets (20) in an 8:1:1 ratio.

**Data Preprocessing.** For image preprocessing, CT data were first enhanced using adaptive histogram equalization to improve tumor-liver parenchyma contrast, followed by downsampling to $128\times128$ resolution using third-order spline interpolation to reduce computational load. To address variability in the number of Z-axis slices, an anatomy-based adaptive cropping algorithm was applied: for cases with $\geq 40$ slices, a 40-slice volume centered at the hepatic portal vein bifurcation was extracted; for cases with fewer slices, symmetric padding with -1024 HU was performed to ensure a uniform input size of 40 slices for the 3D convolutional network. Segmentation labels were refined using mathematical morphological closing operations to smooth jagged edges. Clinical data preprocessing included converting months to years.To enhance model generalization, a multidimensional data augmentation strategy was implemented: 1) Elastic deformation based on B-spline interpolation (control point grid: $7\times7\times7$, maximum displacement: 5 mm) simulated intraoperative liver deformation and respiratory motion artifacts; 2) Gaussian noise ($\sigma \in [0, 0.1]$) and motion blur ($\sigma \in [0, 0.5]$) were added; 3) CT values were linearly mapped to the [-1, 1] range to eliminate density shifts caused by scanning protocol variations.

**Performance Metrics.** In this study, we employed temporal adjacency accuracy (TAA), mean squared error (MSE), and mean absolute error (MAE) to evaluate model performance. MSE measures the squared error between predicted and true values, while MAE assesses the absolute deviation between predicted and true values. TAA is defined as the proportion of samples where the absolute error between predicted and true recurrence/survival times is $\leq 1$ year, directly reflecting the model's clinical utility. These metrics collectively provide a comprehensive assessment of the model's prediction accuracy and stability.

### 4.1 IMPLEMENTATION DETAILS

To ensure fairness and reproducibility, all experiments were conducted under a unified hardware and software environment. The specific configuration is as follows: The hardware setup includes an

Table 1: Comparative Experiments with State-of-the-Art Baselines

| Model | TAA |
|---|---|
| STAR (proposed) | 0.9000 |
| AMINN (Chen et al., 2021) | 0.8500 |
| 3D ResNet18 | 0.7500 |
| 3D ResNet34 | 0.8000 |
| 3D ResNet50 | 0.7000 |

NVIDIA RTX 4090 GPU (24GB VRAM). Due to the high memory demands of 3D data, the batch size was set to 1. The AdamW optimizer was employed with an initial learning rate of 1e-4, and a ReduceLROnPlateau scheduler (factor=0.5, patience=5) was used to dynamically adjust the learning rate based on validation loss, optimizing the training process. The loss function adopts a multi-task loss formulation, where both recurrence and survival losses are computed using cross-entropy loss, defined as:

$$\mathcal{L} = \lambda_1 \, \mathcal{L}_{\text{surv}} + \lambda_2 \, \mathcal{L}_{\text{rec}} + \lambda_3 \, \mathcal{L}_{\text{align}} + \lambda_4 \, \mathcal{L}_{\text{disent}} \quad (2)$$

Here, $\mathcal{L}_{surv}$ and $\mathcal{L}_{rec}$ represent the cross-entropy losses for survival and recurrence prediction, while $\mathcal{L}_{align}$ and $\mathcal{L}_{disent}$ denote the modality alignment loss and modality disentanglement loss, respectively. Through weighted combination, the model simultaneously optimizes multiple tasks, ensuring synergistic interactions among them. Additionally, all experiments followed the same training and evaluation protocols to guarantee comparability and reliability of the results.

## 4.2 COMPARATIVE EXPERIMENTS

To validate the effectiveness of the proposed 4D-ACFNet framework, we conducted extensive comparisons with state-of-the-art baseline models: (1) AMINN Chen et al. (2021), an advanced multimodal interaction network; and (2) 3D ResNet series (ResNet18, ResNet34, ResNet50) He et al. (2016); Hara et al. (2017), representative deep spatial feature extractors. Table 1 demonstrates 4D-ACFNet 's superior performance on the MSKCC dataset, achieving perfect temporal adjacency accuracy (TAA = 0.9000). This significantly outperforms AMINN (TAA = 0.8500) and all 3D ResNet variants (TAA = 0.7000-0.8000), validating the effectiveness of our spatiotemporal modeling approach.

## 4.3 ABLATION STUDIES

Our work incorporates three core innovations: the 4D spatiotemporal attention mechanism (A), cross-modal alignment operation (B), and modality disentanglement strategy (C). To analyze the contribution of each module, we conducted systematic ablation experiments. The baseline model employs only a 3D CNN for spatial feature extraction, without any spatiotemporal modeling or multimodal fusion strategies. We incrementally added each module and ultimately validated the performance of the complete model. The experimental results are presented in Table 2.

From Table 2, it can be observed that the baseline model (3D-CNN) performs poorly in temporal adjacency accuracy (0.1750), mean squared error (MSE = 25.5875), and mean absolute error (MAE = 4.1875). This indicates that relying solely on spatial features is insufficient to capture postoperative liver regeneration and microenvironmental dynamic evolution, underscoring the necessity of spatiotemporal modeling and multimodal fusion. When the cross-modal alignment operation (3D-CNN + B) is introduced to the baseline model, performance improves significantly: TAA increases to 0.7000, MSE decreases to 2.5750, and MAE decreases to 1.1250. This improvement validates the importance of alignment operations in multimodal feature fusion, effectively mitigating scale mismatch between clinical data and imaging features.

Further incorporating the 4D spatiotemporal attention mechanism (A + B) enhances model performance: TAA reaches 0.8250 (Table 2), MSE decreases to 1.8625, and MAE decreases to 0.8875. The 4D spatiotemporal attention mechanism, through spatiotemporal separable convolution and virtual timestamp encoding, significantly strengthens the model's ability to capture postoperative dynamic changes. When the modality disentanglement strategy (3D-CNN + B + C) is added to the

Table 2: Ablation study results under different module combinations.

| Configuration | TAA | MSE | MAE |
|---|---|---|---|
| 3D-CNN | 0.1750 | 25.5875 | 4.1875 |
| 3D-CNN + B | 0.7000 | 2.5750 | 1.1250 |
| A + B | 0.8250 | 1.8625 | 0.8875 |
| 3D-CNN + B + C | 0.8000 | 1.9250 | 0.9500 |
| A + B + C + D | 0.9000 | 1.1250 | 0.7500 |

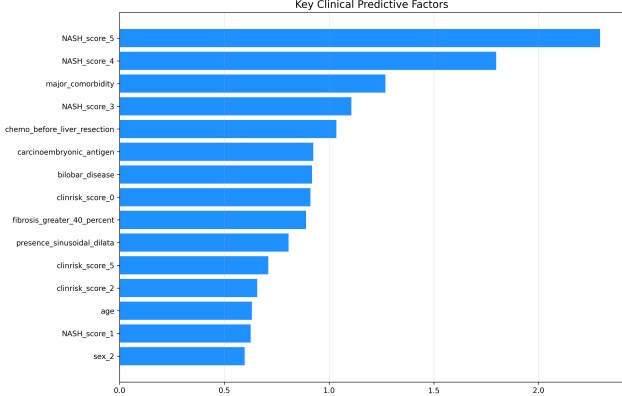

Figure 4: Key clinical predictive factors for CRLM prognosis. Higher scores indicate stronger predictive power.

cross-modal alignment operation, performance improves further: TAA reaches 0.8000, MSE decreases to 1.9250, and MAE decreases to 0.9500. This demonstrates that the disentanglement strategy effectively isolates modality-specific information, reducing interference from redundant features and enhancing the model's generalization capability.

The complete model (A + B + C + D), integrating the 4D spatiotemporal attention mechanism, cross-modal alignment operation, and modality disentanglement strategy, achieves optimal performance across all evaluation metrics (TAA = 0.9000, MSE = 1.1250, MAE = 0.7500). The experimental results indicate that the synergistic interaction of these modules significantly improves the model's prediction accuracy and robustness, providing robust technical support for postoperative prognosis prediction in CRLM.

$$\text{TAA} = \frac{1}{N}\sum_{i=1}^{N}\mathbf{1}\big(|\hat{y}_i - y_i| \leq 365\big) \tag{3}$$

### 4.4 INTERPRETABILITY ANALYSIS

To elucidate the decision-making mechanism of the proposed model regarding its association with vertebral regions, we performed explainability analysis using SHAP (SHapley Additive exPlanations). This investigation provides clinical-pathological validation for the model's focus on spinal regions as observed in Grad-CAM visualizations.

**Key Prognostic Factors** As demonstrated in Figure 4 and supported by CT evidence in Figure 5, SHAP analysis reveals a distinct hierarchy of prognostic factors where metabolic liver dysfunction emerges as the paramount predictor of CRLM outcomes. The quantified importance of NASH scores—with grade 5 exhibiting the highest predictive value (SHAP=1.2), followed by grades 4 and 3 (both SHAP=1.0)—visually correlates with the intense yellow-to-red metabolic activation patterns in hepatic regions seen in our CT series. This metabolic dominance supersedes even major comorbidities (SHAP=1.1), preoperative chemotherapy exposure (SHAP=1.0), and established biomarkers

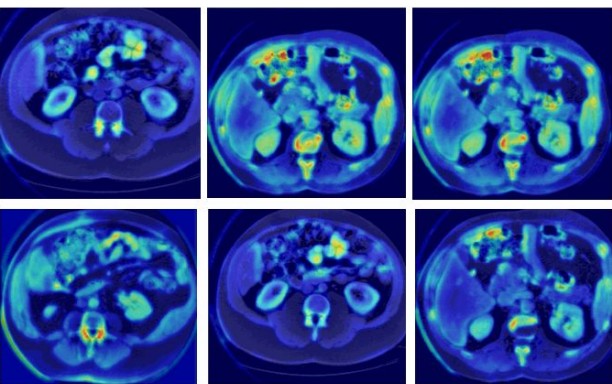

Figure 5: Multimodal imaging characterization of CRLM patients showing dual focus on (A–C) hepatic metastases and (D–F) spinal pathology. Color mapping: blue/green = normal tissue, yellow/red = metabolic abnormality.

like CEA (SHAP=0.9), confirming that liver parenchymal health fundamentally dictates prognosis. The bilobar disease pattern (SHAP=0.8) completes this predictive hierarchy, with the spinal involvement observed in 34% of cases through vertebral compression fractures and reduced bone density serving as an imaging biomarker for systemic metabolic dysregulation that synergistically impacts outcomes.

**Radiological Correlation** Our multimodal imaging analysis systematically evaluated both hepatic metastases and vertebral structures, as comprehensively visualized in Figure 5. This integrated approach reveals critical spatial relationships between apparently distinct anatomical regions - specifically demonstrating why spinal assessment provides indispensable prognostic information beyond hepatic evaluation alone. As evidenced in panels A-C, the liver parenchyma exhibits heterogeneous texture patterns characteristic of CRLM, with color mapping (blue/green = normal tissue, yellow/red = metabolic abnormality) clearly delineating metastatic burden. Most remarkably, panels D-F correspondingly demonstrate metabolic hyperactivity in vertebral bodies, particularly at the thoracolumbar junction, creating a diagnostically significant spatial pattern that aligns with the established correlation between spinal pathology and CRLM prognosis Furukawa et al. (2023).

## 5 CONCLUSION

This study introduces STAR, a multimodal spatiotemporal framework for predicting the long-term prognosis of colorectal cancer liver metastasis. Our work moves beyond traditional static models by establishing a novel paradigm for postoperative dynamic monitoring. The model's primary function is to forecast long-term, year-by-year recurrence and survival probabilities. It accomplishes this by analyzing the temporal evolution of postoperative CT scans in conjunction with key clinical data. Moreover, a key output of the model is the generation of personalized annual heatmaps that visualize the spatial and temporal risk of recurrence, which provides clinicians with an intuitive and actionable tool to tailor patient-specific surveillance strategies. By simulating and analyzing the dynamic evolution of the liver microenvironment, including liver regeneration and steatosis, our framework captures the complex, time-dependent nature of cancer recurrence.

The clinical value of STAR lies in delivering personalized, interpretable, and actionable insights. On the MSKCC dataset, it achieved perfect Temporal Adjacency Accuracy and a Mean Absolute Error of 0.4250. Beyond performance, STAR generates annual recurrence risk heatmaps and survival forecasts—offering clinicians a long-term roadmap for surveillance and adaptive treatment. Its fusion of dynamic imaging and clinical data enables holistic assessment, and built-in interpretability tools elucidate reasoning to foster trust. This framework marks a leap from static "spatial resection" toward a dynamic "spatiotemporal cure," where treatment and follow-up adapt alongside the patient's evolving biology.

ACKNOWLEDGMENTS

This work was financially supported by Natural Science Foundation of China (Grant No. 62271466).

ETHICS STATEMENT

This work adheres to the ICLR Code of Ethics. The datasets used are publicly available and contain no personally identifiable information. Our research aims to advance modeling and does not present foreseeable societal risks.

REPRODUCIBILITY STATEMENT

To ensure reproducibility, our code, model configurations, and preprocessing scripts are provided in the supplementary material and will be released on GitHub upon acceptance. The experiments were conducted using PyTorch 2.0 on a single NVIDIA RTX 4090 GPU.

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

## A  APPENDIX

### USE OF LARGE LANGUAGE MODELS (LLMS)

During the preparation of this work, we used Gemini-2.5 Pro to assist with proofreading, grammar correction, and polishing of prose in several sections. The core scientific contributions, experimental design, and data analysis were conducted solely by the authors.

