# OpenReview forum: "STAR: Spatio-Temporal Attention-guided Recurrence Prediction for Colorectal Cancer Liver Metastasis"
_ICLR.cc/2026/Conference — ICLR 2026 Conference Withdrawn Submission_

### Official Review · Reviewer_fRRw · 2025-10-21

**Soundness:** 1
**Presentation:** 1
**Contribution:** 2
**Rating:** 2
**Confidence:** 4

**Summary:**

This paper introduces STAR (Spatio-Temporal Attention-Guided Recurrence Prediction), a deep learning framework for dynamic prediction of postoperative recurrence and survival in colorectal liver metastasis (CRLM) patients. Unlike conventional static prognostic models, STAR models the temporal evolution of postoperative risk by generating synthetic postoperative imaging sequences from preoperative scans and virtual timestamps. Leveraging a 4D attention mechanism, the model integrates spatio-temporal imaging dynamics with clinical variables to predict year-by-year recurrence risk and survival probability over a 12-year horizon. Furthermore, STAR produces personalized annual recurrence risk heatmaps, providing interpretable and intuitive visual guidance to support dynamic, patient-specific surveillance strategies.

**Strengths:**

### **Strong and clinically grounded motivation**
The study is well-motivated from a clinical perspective, targeting the dynamic prediction of postoperative recurrence risk in colorectal liver metastasis (CRLM) patients. This focus directly addresses a crucial unmet need in personalized follow-up and surveillance planning, demonstrating high practical relevance and originality.
### **Interesting model design**
The paper introduces a novel model architecture that aligns with the task of year-by-year recurrence prediction. The incorporation of 4D spatio-temporal attention, virtual timestamps, and a dual-loss multimodal fusion strategy collectively form an original and technically sound framework. These components are conceptually meaningful for integrating heterogeneous medical data and modeling complex disease progression patterns over time.
### **Interpretability and clinical insight**
The use of SHAP-based interpretability analysis, along with correlation studies linking model explanations to prior clinical knowledge, enhances both transparency and credibility. The personalized, annual recurrence risk heatmaps provide intuitive visualization that can assist clinicians in understanding and validating the model’s predictions, thereby increasing the potential clinical impact of the work.

**Weaknesses:**

### **Misalignment between the abstract and the actual implementation**
The abstract and introduction emphasize modeling the temporal evolution of postoperative CT scans, but the experimental data only include preoperative images. The postoperative images are synthetically generated using virtual timestamps, yet there is no real postoperative data to validate the quality or fidelity of these generated scans. This inconsistency undermines the credibility of the proposed framework.

**Suggestion:** The authors should clarify this conceptual discrepancy and, ideally, provide validation of the generated postoperative images—either through a small-scale real postoperative dataset or expert evaluation—to substantiate the claim of modeling temporal evolution.

### **Limited dataset size and lack of external validation**
The dataset used in the study contains fewer than 200 samples, with only 20 in the test set, and there is no external or cross-task validation. Such a small and homogeneous sample severely limits the model’s generalizability and robustness, especially for a clinical prediction task.

**Suggestion:** The authors are encouraged to perform external validation on an independent dataset or at least apply cross-validation with stratified folds.

### **Insufficient comparison with recent state-of-the-art 4D architectures**
The paper only compares the proposed model with a single 2021 baseline, neglecting recent 4D spatio-temporal architectures, especially transformer-based models such as 4D Swin-ViT or TimeSformer. Given that STAR relies heavily on 4D attention and temporal modeling, the absence of these comparisons raises concerns about the rigor and completeness of the experimental evaluation.

**Suggestion:** Including comparisons with representative modern 4D attention-based models—or, at minimum, a detailed discussion justifying their exclusion—would significantly strengthen the technical evaluation.

### **Lack of clarity and inconsistencies in presentation**
The paper suffers from clarity and organization issues that hinder reproducibility and comprehension. Specifically:
1. The Related Work section lacks structured organization and fails to differentiate between temporal modeling, multimodal fusion, and interpretability research.
2. The mathematical description of virtual timestamp embedding and the dual loss formulation is abstract and incomplete, lacking sufficient detail for replication.
3. There are figure-text mismatches (e.g., line 253 refers to Figure 3 content that does not align with the figure).
4. The “4D-ACFNet” mentioned in Section 4.2 does not appear in the methods section.

**Suggestion:** The authors should revise for consistency, include complete mathematical definitions, and reorganize the related work into thematic subsections to improve readability and technical clarity.

**Addressing these concerns would substantially strengthen the paper. I will reconsider my overall score if the authors can adequately address the concerns raised above.**

**Questions:**

**Handling of Irregular and Missing Follow-up Data**

In real-world clinical scenarios, longitudinal follow-up data are often incomplete, irregularly sampled, or contain missing time points due to practical constraints in patient monitoring. Could the authors clarify whether their proposed model is robust to such data irregularities? Specifically, how would the model perform if the time intervals between follow-ups are non-uniform or if certain temporal data are missing?
- It would be helpful to discuss whether the model incorporates any temporal alignment, imputation, or interpolation strategies, or if it relies on uniform time sampling during training and inference.
- If the current approach assumes complete or regularly spaced data, the authors might consider discussing possible extensions or adaptations (e.g., continuous-time modeling, neural ODEs, or masked temporal attention) that could make the model more applicable to real-world longitudinal datasets.

**Interpretation of Postoperative Microenvironmental Dynamics**

The paper suggests that the model captures postoperative microenvironmental changes or disease progression patterns over time. Could the authors elaborate on what specific patterns or regularities were observed?
- For instance, does the model reveal consistent temporal trajectories or phenotypic shifts associated with recovery or recurrence?
- Are there any biologically interpretable trends emerging from the learned representations (e.g., clustering of similar progression profiles, correlations with clinical outcomes)?
Providing such interpretability or biological insight would strengthen the paper’s contribution, as it would demonstrate that the model not only achieves predictive performance but also yields clinically meaningful understanding of disease evolution.

---

### Official Review · Reviewer_E7QS · 2025-10-31

**Soundness:** 2
**Presentation:** 2
**Contribution:** 1
**Rating:** 0
**Confidence:** 5

**Summary:**

The paper proposes STAR (Spatio‑Temporal Attention‑Guided Recurrence Prediction), a deep‑learning framework for the long‑term postoperative management of colorectal cancer liver metastasis (CRLM).

**Strengths:**

1.The clinical task (cancer recurrence) has great significance in health care.

2. The author combined imaging data and clinical variables for multi-modal prognosis.

3. The author visualized the risk regions and enhanced the model's explainability.

**Weaknesses:**

1. Limited novelty. Previous works with similar architecture of spatiotemporal attention has been proposed. For example
Temporal Context Matters: Enhancing Single Image Prediction with Disease Progression Representations (CVPR 2022)
Multimodal Spatiotemporal Graph Neural Networks for Improved Prediction of 30‑day All‑Cause Hospital Readmission (JBHI 2023)


2. The model is trained and tested on just 197 patients from a single institution. The metrics in Table 1 are reported to four decimal places, overstating the precision achievable from such a small dataset.

3. Lack of baseline comparisons.

**Questions:**

1.How many patients actually have 12‑year longitudinal scans? How do you handle missing years and censoring?
2.Can you report standard survival metrics (C‑index, time‑dependent AUC) and compare against established survival models?

---

### Official Review · Reviewer_nTGF · 2025-11-01

**Soundness:** 1
**Presentation:** 1
**Contribution:** 1
**Rating:** 2
**Confidence:** 4

**Summary:**

This paper presents STAR, a deep learning framework for dynamic postoperative prognosis in colorectal cancer liver metastasis (CRLM). Unlike static prognostic tools, STAR integrates temporal CT imaging and clinical data to predict year-by-year recurrence and survival risk. It produces personalized, spatial-temporal recurrence heatmaps to enhance interpretability and guide surveillance. Evaluated on the MSKCC CRLM dataset, STAR achieved 90% temporal adjacency accuracy and an MAE of 0.75 over 12 years. Overall, this work enables continuous monitoring of colorectal cancer recurrence, demonstrating potential for personalized postoperative management through interpretable spatio-temporal modeling.

**Strengths:**

[S1] Clinically meaningful task with interpretable validation:
This paper addresses a clinically meaningful problem—predicting colorectal cancer recurrence in CRLM patients. Beyond demonstrating predictive performance, the authors strengthen their study with interpretability analyses, including SHAP-based feature attribution and Grad-CAM visualizations, to link the model’s learned representations with established clinical knowledge. These analyses provide more insights into the model’s decision process and its alignment with known medical evidence.

**Weaknesses:**

While the paper presents a clinically valuable application, several aspects of the methodology and evaluation remain underdeveloped or insufficiently clarified.

[W1] Ambiguity in target definition and causal design:
The yearly recurrence labels and post-recurrence prediction process are not clearly defined. It is also unclear whether STAR includes any causal design elements or a temporal masking strategy to prevent future information leakage during model training or inference, which is critical for time-dependent medical prediction tasks.

[W2] Insufficient methodological clarity:
Several key modules in Algorithm 1 and Figure 2(d) lack precise descriptions. For instance, the roles of the Conv3D block, h_recon, and the gating variables (gate_0, gate_1) are not well explained, and the purpose of the L_align loss (enforcing dissimilarity between reconstructed and original CTs) appears counterintuitive. These gaps make it difficult to assess the soundness and interpretability of the model design.

[W3] Incomplete and potentially unreliable quantitative evaluation:
Table 1 omits MSE and MAE results, and the small test size (20 samples) raises concerns about the robustness of the results. The lack of confidence intervals, statistical significance testing, or bootstrapping analysis makes it hard to judge whether STAR’s reported superiority is statistically meaningful.

[W4] Limited experimental scope and missing baselines:
The comparison set excludes several relevant and recent spatio-temporal or CRLM-related architectures (e.g., STG, Temporal Attention Unit). Without these baselines, it is difficult to contextualize STAR’s claimed advantages relative to state-of-the-art spatio-temporal modeling methods.

[W5] Presentation and referencing issues:
Several minor clarity issues persist, including missing or duplicated abbreviations, a lack of references for the STG framework, insufficient detail in Figure 3 (e.g., mask color interpretation and CT axis orientation), and the omission of details regarding key clinical features used in the SHAP analysis.

**Questions:**

[Q1] Definition of yearly targets and causality:
Please clarify how the yearly recurrence labels are defined, particularly in cases where recurrence has already occurred or when no recurrence is observed during follow-up. Additionally, does STAR incorporate any causal mechanism to prevent information leakage from future time points during prediction?

[Q2] Use of h_recon and reconstruction Loss:
In Algorithm 1, the purpose of h_recon is ambiguous since Figure 2(d) already outputs spatio-temporal CT representations. Also, is there a reconstruction loss associated with STAR? If not, can the deconvolved outputs in Figure 2(d) still be considered CT reconstructions?

[Q3] Clarification for the gating and alignment mechanisms:
The process for obtaining gate_0 and gate_1 in Algorithm 1 is not described. Please specify how these gating signals are computed and their roles in the learning dynamics. Also, in Algorithm 1, enforcing reconstructed CTs to be “dissimilar” from the originals by L_align seems counterintuitive. Clarification is needed on the rationale and intended behavior of this loss term.

[Q4] Incomplete quantitative and statistical results:
The results for MSE and MAE are not provided in Table 1. Including them is important for a complete performance comparison. Also, given the small test set (n = 20), please include statistical analyses (e.g., 95% confidence intervals via bootstrapping, significance testing between top models) to support claims of superiority and robustness.

[Q5] Comparisons with recent models:
It would strengthen the evaluation to include more recent or relevant baselines, such as CRLM-specific architectures (e.g., STG, mentioned in line 127) or efficient spatio-temporal attention models (e.g., Temporal Attention Unit (CVPR 2023)).

---

### Official Review · Reviewer_9hc3 · 2025-11-02

**Soundness:** 2
**Presentation:** 2
**Contribution:** 2
**Rating:** 4
**Confidence:** 4

**Summary:**

The paper proposes STAR, a spatiotemporal multimodal framework that combines CT imaging with clinical variables to forecast year-by-year postoperative recurrence and survival for CRLM patients. A tracker uses separable 3D/temporal convs with 4D attention and a virtual timestamp to model dynamics; a fusion module aligns clinical and imaging features; and a projector outputs annual risks and heatmaps. On a 197-patient MSKCC cohort, the paper reports strong TAA and MAE along with ablations indicating each module helps. Figures and tables summarize the architecture and reported gains.

**Strengths:**

- Ambitious clinical goal: per-year risk trajectories and spatial maps intended to guide follow-up
- Clear high-level pipeline tying imaging dynamics to clinical variables
- Ablations suggest each component contributes

**Weaknesses:**

- I find a disconnect between the claimed dynamics and the actual data. The method is described as modeling postoperative spatiotemporal evolution, but the dataset seems to consist of preoperative CT scans with survival labels. The tracker’s use of a simulated timestamp makes me question whether true longitudinal information or recurrence-location labels exist. Without that, I have trouble trusting that the spatial heatmaps or temporal dynamics reflect postoperative changes.
- The survival analysis methodology and choice of baselines raise concerns for me. The authors train with cross-entropy and evaluate with TAA/MAE but don’t handle censoring or use standard survival metrics such as C-index, time-dependent AUC, or calibration. The baselines also omit widely used survival models (CoxPH, RSF, DeepSurv), which makes it difficult for me to judge the improvement. I’m also not clear on how censored samples are treated in training or evaluation.
- Nit: The paper’s internal consistency and reporting issues undermine my confidence. There are mismatched names (STAR vs. 4D-ACFNet), unexplained module D in the ablation table, MAE values differ between abstract and conclusion, etc. The interpretability analysis focuses on vertebral metabolic features and NASH grades that don’t appear to be part of the dataset description.

**Questions:**

Please address the concerns raised above.

---

### Note · Authors · 2025-12-06

**Comment:**

I have read and agree with the venue's withdrawal policy on behalf of myself and my co-authors.

**Withdrawal Confirmation:**

I have read and agree with the venue's withdrawal policy on behalf of myself and my co-authors.